

# In-situ parameters, nutrients and dissolved carbon distribution in the water column and pore waters of Arctic fjords (Western Spitsbergen) during a melting season

Seyed Reza Saghravani[1,*], Michael Ernst Böttcher[2,3,4], Wei-Li Hong[5,6], Karol Kuliński[1], Aivo Lepland[7],

Arunima Sen[8,9], Beata Szymczycha[1]

[1]Marine Chemistry and Biochemistry Department, Institute of Oceanology Polish Academy of Sciences (IOPAN), Powstańców Warszawy 55, Sopot 81-712, Poland
[2]Geochemistry and Isotope Biogeochemistry, Leibniz Institute for Baltic Sea Research (IOW), Seestrasse 15, D-18119
Warnemünde, Germany
[3]Marine Geochemistry, University of Greifswald, D-17489 Greifswald, Germany
[4]Maritime Systems, Interdisciplinary Faculty, University of Rostock, D-18059 Rostock, Germany
[5]Department of Geological Sciences, Stockholm University, Svante Arrhenius väg 8, Stockholm 11418, Sweden
[6]Baltic Sea Centre, Stockholm University, Universitetsvägen 10 A, 10691, Stockholm, Sweden
[7]Geological Survey of Norway, Leiv Eirikssons vei 39, 7040, Trondheim, Norway
[8]Department of Arctic Biology, University Centre in Svalbard, N-9171, Longyearbyen, Norway
[9]Faculty of Bioscience and Aquaculture, Nord University, 8049, Bodø, Norway

*Correspondence to: Seyed Reza Saghravani (reza@iopan.pl)


**Abstract.** A nutrient distribution such as phosphate ($PO_4^{3-}$), ammonium ($NH_4^+$), nitrate ($NO_3^-$), dissolved silica (Si), total dissolved nitrogen (TN), dissolved organic nitrogen (DON) together with dissolved organic carbon (DOC) and inorganic carbon (DIC), was investigated during a high melting season in 2021 in the western Spitsbergen fjords (Hornsund, Isfjorden, Kongsfjorden and Krossfjorden). Both the water column and the pore water were investigated for nutrients and dissolved

carbon distribution and gradients. The water column concentrations of most measured parameters such as $PO_4^{3-}$, $NH_4^+$, $NO_3^-$, Si, and DIC showed significant changes among fjords and water masses. In addition, pore water gradients of $PO_4^{3-}$, $NH_4^+$, $NO_3^-$, Si, DIC and DOC revealed significant variability between fjords and are likely substantial sources of the investigated elements for the water column. The obtained dataset reflects differences in hydrography and biogeochemical ecosystem function of the western Spitsbergen fjords and may form the base for further modelling of physical oceanographic and

biogeochemical processes within the investigated fjord systems. All data described in this paper are stored in the Zenodo online repository; https://doi.org/10.5281/zenodo.10523197 (Szymczycha et al., 2024).





## 1 Introduction

The Arctic is facing significant and rapid transformations due to Arctic amplification accelerating climate change in the region (Dunse et al., 2021). Warming of climate causes changes in oceanic and atmospheric circulation patterns, permafrost degradation, a decrease in the thickness and extent of sea ice, as well as a shrinkage of glaciers (IPCC, 2019; Dunse et al., 2021). Freshwater released from glacial meltwater runoff or frontal ablation and accompanied fluxes of solutes, is a significant factor that changes the hydrographic pattern and biogeochemistry of water masses, which in turn affects the biological

productivity in the ocean and fjords (Hopwood et al., 2016, 2020).

Many studies have investigated the biogeochemistry of nutrients in the Barents Sea and Arctic region (Henley et al., 2020; Gundersen et al., 2022; Tuerena et al., 2022). Substantial efforts have been made in existing Arctic monitoring programmes, research initiatives, and scientific projects to describe, explain and predict environmental changes due to diverse pressures for the Arctic ecosystem (Townhill et al., 2022). Studies indicate that net primary production in open Arctic waters is mainly

sustained by the upwelling of nutrients and light availability (Henley et al., 2020; Stroeve et al., 2021) while nitrogen is considered to be the key limiting nutrient in the Arctic Ocean (Mills et al. 2018; Ko et al. 2020). In addition, Henley et al. (2020) indicated that, with ongoing sea ice losses and Atlantification, the expected shift from more Arctic-like ice-impacted conditions to more Atlantic-like ice-free conditions is expected to increase nutrient availability and the duration of the vegetation period in the Arctic shelf region.

Arctic fjords have not gained similar attention and investigations were usually taken in individual fjord systems (Codispoti et al., 2013; Henley et al., 2020; Kim et al., 2020; Pogojeva et al., 2022). Spatially wide studies of fjords and investigations focusing on the hydrography and biogeochemical functioning of the Arctic shelf seafloor are still lacking. To address the existing knowledge gaps, we studied the water masses, and pore waters, together with their biogeochemical composition in the western Spitsbergen fjords. The selected area is an excellent research site for investigating the effects of both rapidly

occurring climate change and varied levels of Atlantification, as different fjords are under the diverse impact of the East Spitsbergen Current bringing cold Arctic Water (ArW) and the West Spitsbergen Current carrying warmer and more saline Atlantic Water (AtW). This was also our motivation to release this macronutrient dataset, which we believe may constitute a biogeochemical reference for other experimental and modelling research in the region.

## 2 Materials and methods

### 2.1 Study area description

The west coast of the Svalbard archipelago (76 - 80°N) consists of different fjords and sub-fjords (Fig. 1). All investigated fjords (Hornsund, Isfjorden, Kongsfjorden and Krossfjorden) are influenced by the East Spitsbergen Current carrying cold

Arctic Water (ArW) from the Barents Sea and the West Spitsbergen Current with warmer and more saline Atlantic Water (AtW) from the Norwegian Sea (Promińska et al., 2018) (Fig. 1). When AtW mixes with ArW, the warmer Transformed Atlantic Water (TAW) forms (Cottier et al., 2005). Surface water (SW) is formed locally from glacial melt and river runoff,





and occupies the surface layer of the fjord. Intermediate Water (IW) forms as a result of mixing AtW or TAW with overlying fresher SW. Local Water (LW) and Winter Cooled Water (WCW) forms usually during autumn and winter (Cottier et al.,

2005; Hop et al., 2006; Cantoni et al., 2020) in depressions within the inner fjords.

Hornsund is located at the southern end of Spitsbergen, is about 30km long and about 15km wide. The fjord is divided into the main basin and inner basin (Brepollen) by a shallow sill located in the centre of the fjord (Błaszczyk et al., 2019). The average depth is approximately 90m, while the deepest reaches 250m (Moskalik et al., 2014). Sediments consist of mud and sandy mud, laminated mud, homogeneous to bioturbated mud and sandy gravel (Drewnik et al., 2016). Freshwater discharge to the

fjord was estimated to be approximately 1.8km$^3$ annually (Weslawski et al., 1991), mainly due to glacier melting (64%) with the fastest retreating rate in Svalbard (with an average rate between 100 and >200m·yr$^{-1}$ (Grabiec et al., 2018). Other freshwater sources, such as frontal ablation and river runoff, influence primarily the upper water column (Zaborska et al., 2020). Hornsund exhibits high nutrient enrichment and experiences a strong influence from the ArW and colder coastal water (Włodarska-Kowalczuk et al., 1998). These conditions contribute to greater productivity in Hornsund compared to the warmer and saline

fjords (Santos-Garcia et al., 2022).

Isfjorden stands as the largest fjord system on Spitsbergen having about 100km length from the mouth to the head and up to 425m deep. Isfjorden has several subfjords and bays. Studies conducted in Isfjorden have provided evidence of the significant impact of freshwater on the water column (McGovern et al., 2020; Finne et al., 2022). Seasonal stratification has been responsible for the retention of terrestrial carbon and nutrients within the euphotic zone and a decrease in vertical mixing

during the most productive season (McGovern et al., 2020; Finne et al., 2022). The enhanced freshwater input contributes to the overall nutrient loading in the system, affecting the biogeochemical processes and ecosystem functioning.

Krossfjorden exhibits a south-east to north-west orientation, stretching approximately 30km in length and reaching widths from 3km to 6km. The total volume of Krossfjorden 25km$^3$ and a maximum depth of 373m (Svendsen et al., 1992). Krossfjorden, characterized by a colder spring and less intrusion of AtW, shares similar conditions to the inner part of

Kongsfjorden. However, it experiences a shorter period of glacier retreat compared to Kongsfjorden (Gamboa-Sojo et al., 2022). Studies on chlorophyll and other pigment distribution in surface sediments suggest that Krossfjorden is more productive than Kongsfjorden (Singh and Krishnan, 2019).

Kongsfjorden is around 20km long and up to 10km wide with an orientation from south-east to north-west (Promińska et al., 2017). The depth at the mouth of the fjord is about 360m and decreases toward the inner part where does not exceeds 100m

(Svendsen et al., 2002). Kongsfjorden has remained sea ice-free since 2011, invoking profound biogeochemical transformations (Hop and Wiencke, 2019; Pavlova et al., 2019). Unlike other Arctic fjords, it experiences a distinct influence from the intrusion of warm and saline waters (Hodal et al., 2012). The inflow of AtW and ArW from one side and glacier meltwater from another (Halbach et al., 2019) lead to both amplified nutrients and carbon cycling, enhanced net primary productivity and oxygen depletion in deeper waters (Santos-Garcia et al., 2022).




## 2.2 Sampling and Analyses

Sampling was carried out from 25$^{th}$ July to 20$^{th}$ August 2021 on board the r/v Oceania belonging to the Institute of Oceanology, Polish Academy of Sciences (IOPAN). A towed CTD profiling system (rosette) equipped with 10L Niskin bottles was used to collect water samples from 3 to 5 depths at each location (selected based on salinity and oxygen profiles). Temperature (T),
salinity (S) and oxygen ($O_2$) concentration were measured in situ using a Sea-Bird Scientific SBE 911 Plus CTD profiler (calibrated prior to the cruise). Seawater pH was measured with a WTW Multi 3400i Multi-Parameter Field Metre that yielded an accuracy of 0.1. The pH results are given at 25°C. GEMAX and Nemisto gravity corers were used to collect up to approximately 40cm long sediment cores.

### 2.2.1 Seawater sampling

Seawater pH was measured 10ml of seawater was filtered (cellulose acetate filters with a pore size of 0.45μm), frozen in a pre-cleaned high-density polyethene bottle and stored at -20°C for further nutrient analysis. The seawater for DIC analysis was transferred into the pre-cleaned 250ml glass bottle and poisoned with 100μl saturated $HgCl_2$. 20 ml of seawater for DOC and TN analysis were filtered through 0.45μm MN GF-5 filters and transferred into the pre-combusted glass bottle and acidified
to pH~2 with $HCl_{conc.}$ to stop mineralization and remove carbonates.

### 2.2.2 Pore water sampling

Pore water was extracted from sediments via Rhizon® samplers (Rhizosphere, diameter of 2.5mm, and mean pore size of 0.15μm) directly after extracting the cores. Up to 5ml of pore water was frozen in a pre-cleaned high-density polyethene bottle
and stored at -20°C for further nutrient analysis and approximately 2ml of pore water was kept in PE vials for further $Cl^-$ analysis. 12ml of pore water was transferred into the pre-combusted glass bottle and poisoned with $HgCl_2$ for further DIC, DOC and TN analysis.

### 2.2.3 Chemical Analyses
Nutrient concentrations were determined using a SEAL AA500 AutoAnalyzer (Seal Analytical) applying standard photometric methods (Grasshof et al., 1983). Quality control consists of repeated measurements of two different CRMs (QC3179, Sigma Aldrich and HAMIL, Environment Canada). Method detection limits are 0.33μmol L$^{-1}$ for nitrate ($NO_3^-$), 0.27μmol L$^{-1}$ for $NH_4^+$, 0.1μmol L$^{-1}$ for phosphate ($PO_4^{3-}$), 0.3 μmol L$^{-1}$ for dissolved silicates (Si). The accuracy of $NO_3^-$, $NH_4^+$, $PO_4^{3-}$ and Si measurements was 98.8%, 98.8%, 99.0% and 100.1%, respectively, while the precision was 0.01μmol L$^{-1}$, 0.02μmol L$^{-1}$,
0.01μmol L$^{-1}$ and 0.03μmol L$^{-1}$, respectively. Chloride ($Cl^-$) was determined by titration (Mohr's Method; Belcher et al., 1957). The DIC analyses were carried out based on sample acidification with Apollo SciTech's AS-C6L DIC Analyzer equipped with the laser-based $CO_2$ detector (LI-7815, Li-Cor, USA). The accuracy for DIC measurements was ensured by using certified reference materials (CRMs, batches no. #190 and #195) from A.G. Dickson (Scripps Institution of Oceanography, USA) and the precision was obtained from triplicate measurements of individual samples and was not worse than ± 3 μmol L$^{-1}$ with an





average recovery 99.0%. The DOC and TN analyses were done in a TOC-L analyzer (Shimadzu) using a high temperature (680°C) oxidation method with Pt catalyst. The precision of the DOC measurements was $\pm$ 4µmol $L^{-1}$ ; the accuracy was determined by repeated measurements of the certified reference materials (CRMs) provided by the D. Hansell Laboratory (University of Miami, USA), the recovery was 99%. The accuracy of the TN measurements was guaranteed by using the same CRMs used to determine DOC, average recovery was 97%. DON was determined by subtracting the sum of $NO_3^-$ and $NH_4^+$

from TN results.

### 3 Data description

### 3.1 Water masses distribution

Different water masses were distinguished within the investigated fjords (Fig.2; Szymczycha et al., 2024). The classification

was done based on Cottier et al., (2005), Nilsen et al., (2008) and Promińska et al., (2018) separately for each fjord. All the identified water masses align with those previously recognised in Arctic regions (Rudels et al., 2000). However, some interesting differences have been found between the fjords. In Hornsund SW, ArW, WCW and IW were found. In Isfjorden SW, ArW, IW, LW and TAW occurred. In Kongsfjorden and Krossfjorden IW, TAW and AT were observed. It is worth noticing that Hornsund did not show any impact by TAW and AtW.


### 3.2 Water column data

The distribution of T, S, pH, $O_2$, $NO_3^-$, $NH_4^+$, $PO_4^{3-}$, Si, DIC, DOC, TN and DON in summer 2021 in western Spitsbergen fjords was investigated. The obtained results were divided into the fjords such as Hornsund (marked blue), Isfjorden (marked grey), Kongsfjorden (marked red) and Krossfjorden (marked grey) (Fig.3; Szymczycha et al., 2024). In all studied fjords

similar trends were observed such as decrease of T, pH and increase of S, $NO_3^-$, $NH_4^+$, $PO_4^{3-}$, Si, DIC, and TN with depth, while, $O_2$, DOC and DON were variable with depth and did not show any pattern. To show the variability of measured parameters between fjords and separate the most freshened surface waters, the results were divided into the surface water layer (the uppermost layer up to 5m: based on salinity and temperature) and the bottom water (the lowermost layer in the water column) in each fjord (Fig. 4). Generally, the temperature of the surface water was warmer than that of the bottom water and

shows a significant difference between the fjords (p=0.00005) with the highest in Isfjorden and the coldest in Kongsfjorden. The bottom water temperature was similar in all fjords (p=0.1732), however only in Hornsund reached negative values. Salinity was much higher in bottom water (median>34) than in surface water (median<33.5) and did not show significant differences between fjords (p<0.05) in both surface and bottom water. The pH (at 25°C) of the surface water was high (median>7.8) and did not vary significantly between the fjords, while the pH of the bottom water was lower than the pH of the surface water and

differed significantly between the fjords (p=0.0109). The median concentration of $O_2$ in both surface and bottom water was comparable and ranged from 308.8µmol $L^{-1}$ to 333.8µmol $L^{-1}$. $NO_3^-$, $NH_4^+$, $PO_4^{3-}$ and DIC showed a significant difference in the median concertation between surface and bottom water and significantly varied between fjords in both water types (p<0.05). Si showed a pattern similar to that of the other nutrients; however, in Isfjorden no significant change was observed





between the surface and bottom. DOC did not change substantially between fjords and between water types. Interestingly,
DOC, TN and DON showed similar behavior in all fjords.

### 3.3 Biogeochemistry of the water masses

In general, all fjord systems are transition zones between land and sea, resulting in complex and dynamic environments
(Schlegel et al., 2023). The West Spitsbergen Fjords are highly stratified basins (Fig. 3) and provide a pathway for the exchange
of heat, salt, nutrients, and dissolved carbon between near-glacier waters and adjacent coastal regions (Hopwood et al., 2020).
These coastal regions are additionally under varying influence of the East Spitsbergen Current and the West Spitsbergen
Current (WSC), which brings cold ArW and warmer and more saline AtW, respectively. It is worth mentioning that WSC in
addition to transporting the majority of heat also transports carbon and plankton supply (Menze et al., 2020). However, WSC
along its way up to Kongsfjorden is being depleted in nutrients (Smoła, 2017) and therefore the influence on Kongsfjorden
will be different from that of the Isfjorden. Thus, understanding the biogeochemical processes in the fjords and characterizing
the differences among them is not possible without a detailed understanding of the water circulation. To characterize the
distribution of T, S, pH, $O_2$, $NO_3^-$, $NH_4^+$, $PO_4^{3-}$, Si, DIC, DOC, TN and DON in the investigated fjords, we used the Kruskal-
Wallis test to characterize the differences in the concentrations of these constituents between different water masses within and
between investigated fjords (Fig. 5). The p-value is presented only if there was a significant difference in the median
concentration of the parameter considered between the investigated fjords. SW, ArW and IW were characterized with different
composition of most of the measured parameters such as $NO_3^-$, $NH_4^+$, $PO_4^{3-}$, Si, and DIC between fjords. Besides $NH_4^+$, Arctic
Water is enriched in nutrients and DIC in Hornsund in comparison to Isfjorden. However, the LW, which was only observed
in Isfjorden, was characterized by the highest concentration of $NO_3^-$, $PO_4^{3-}$, Si, and DIC between all water masses.

### 3.4 Pore water data

The distribution and gradients of $Cl^-$, $NO_3^-$, $NH_4^+$, $PO_4^{3-}$, Si, DIC, DOC in pore waters in the investigated fjords are presented
in Figure 6 (Szymczycha et al., 2024). Generally, $Cl^-$, $NH_4^+$, $PO_4^{3-}$, Si, and DIC increased with depth and $NO_3^-$ and DOC
decreased with depth. $PO_4^{3-}$ decreased in every fjord except Isfjorden. To verify whether sediments are a source of chemical
species in the water column, the concentrations of investigated parameters in pore water up to 5 cm and the concentrations in
bottom water were compared in Figure 7. The median concentrations of $Cl^-$ in pore water did not differ significantly among
fjords and were comparable to those of bottom water, apart from Isfjorden, where the median concentrations of $Cl^-$ in pore
water were smaller than those of bottom water. In all fjords, $NO_3^-$ was higher in bottom water compared to pore water, while
$NH_4^+$, $PO_4^{3-}$, Si, and DIC were significantly higher in pore water in comparison to bottom water. The median concentration of
$NO_3^-$, $NH_4^+$, $PO_4^{3-}$, Si, and DIC was significantly different in both water types (p<0.05). The median concentration of DOC
was slightly higher in pore water than in bottom water. However, it is worth noticing that the concentration ranges for all of
the measured parameters differ between and within the investigated fjords.



## 4. Applications of the dataset

This dataset is beneficial for the broad scientific community that is interested in arctic physical oceanography and marine
biogeochemistry. In addition, dataset sheds light on the spatial distribution of nutrients and the dissolved carbon in the Arctic
fjords. The data presented here are made accessible in the belief that their wide dissemination will lead to greater understanding
and new scientific insights into the nutrient cycles in the Arctic fjords. Possible applications can include: 1) being a reference
and allowing comparison of the future measurements of the nutrients and dissolved carbon distribution in both the water
column and sediments in the same region, 2) determining the C:N:P:Si ratios in different water masses and their comparison
between fjords, to assess the environmental controls and limiting factors for the primary production, and 3) parameterization,
validation, and improvement of existing and future biogeochemical models.

## 5. Data Availability

All data described in this paper are stored in the Zenodo online repository (https://doi.org/10.5281/zenodo.10523197
(Szymczycha et al., 2024)).



## 6. List of figures:

**Fig.1** Study Area including the general map of Spitsbergen, highlighting the depths of the fjords and the surrounding Svalbard
shelf. The warm West Spitsbergen Current and cold East Spitsbergen Current are indicated by red and blue arrows,
respectively. Study sites located in Isfjorden, Kongsfjorden, Krossfjorden, and Hornsund are presented as black triangles.



**Fig.2** The water masses distribution such as surface water (SW), Arctic Water (ArW), Winter-Cooled Water (WCW), Intermediate Water (IW), Local Water (LW), Transformed Atlantic Water (TAW) and Atlantic Water (AtW) in the in Hornsund, Isfjorden, Kongsfjorden and Krossfjorden.



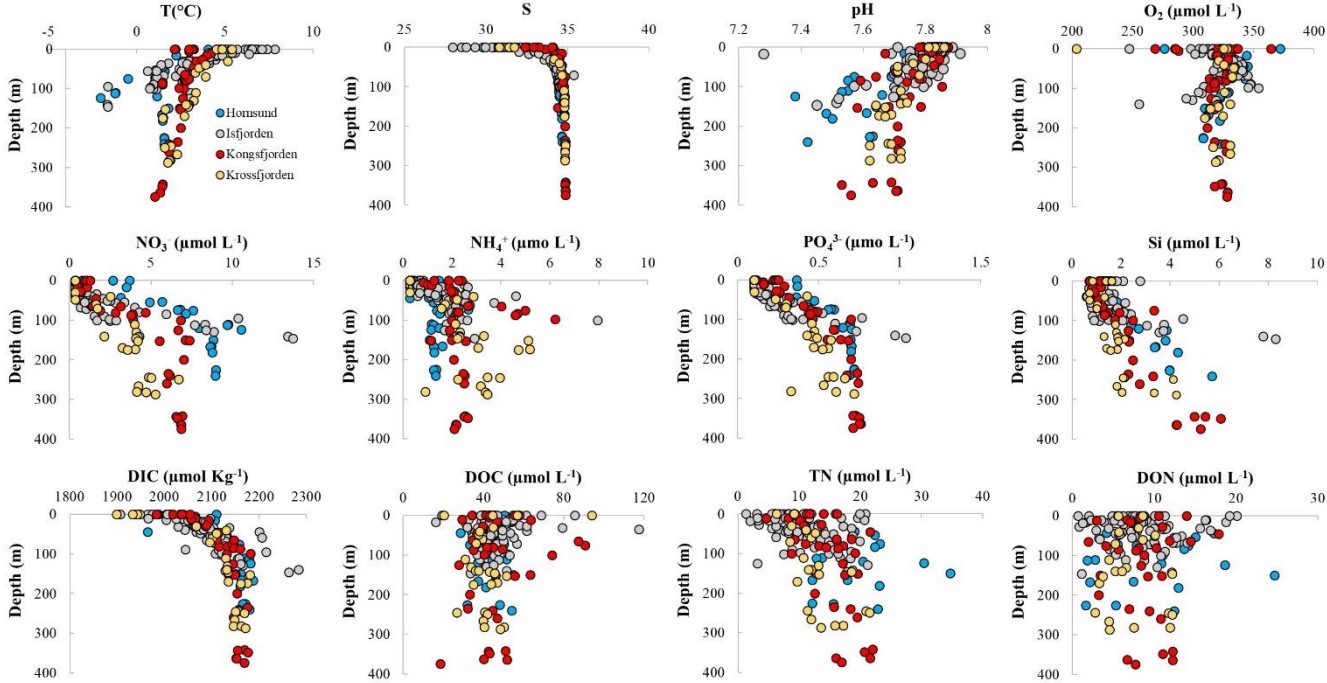

**Fig.3** Distribution of temperature (T), salinity (S), pH, oxygen ($O_2$), nitrate ($NO_3^-$), ammonium ($NH_4^+$), phosphate ($PO_4^{3-}$), dissolved silica (Si), dissolved inorganic carbon (DIC), dissolved organic carbon (DOC), total dissolved nitrogen (TN) and dissolved organic nitrogen (DON) in Hornsund (marked blue), Isfjorden (marked grey), Kongsfjorden (marked red) and Krossfjorden (marked grey).



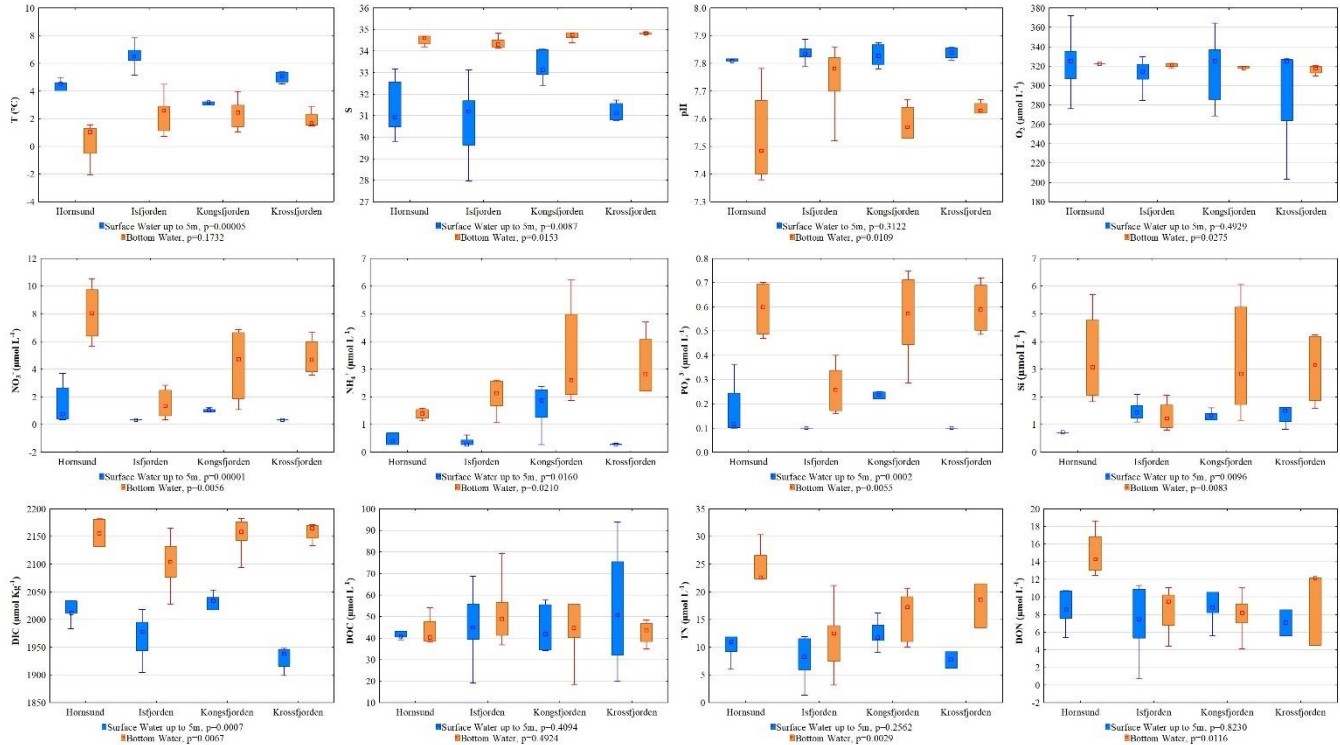

**Fig.4** Box plots of temperature (T), salinity (S), pH, oxygen ($O_2$), nitrate ($NO_3^-$), ammonium ($NH_4^+$), phosphate ($PO_4^{3-}$), dissolved silica (Si), dissolved inorganic carbon (DIC), dissolved organic carbon (DOC), total dissolved nitrogen (TN) and dissolved organic nitrogen (DON) in surface water (marked blue) and bottom water (marked orange) in Hornsund, Isfjorden, Kongsfjorden and Krossfjorden.

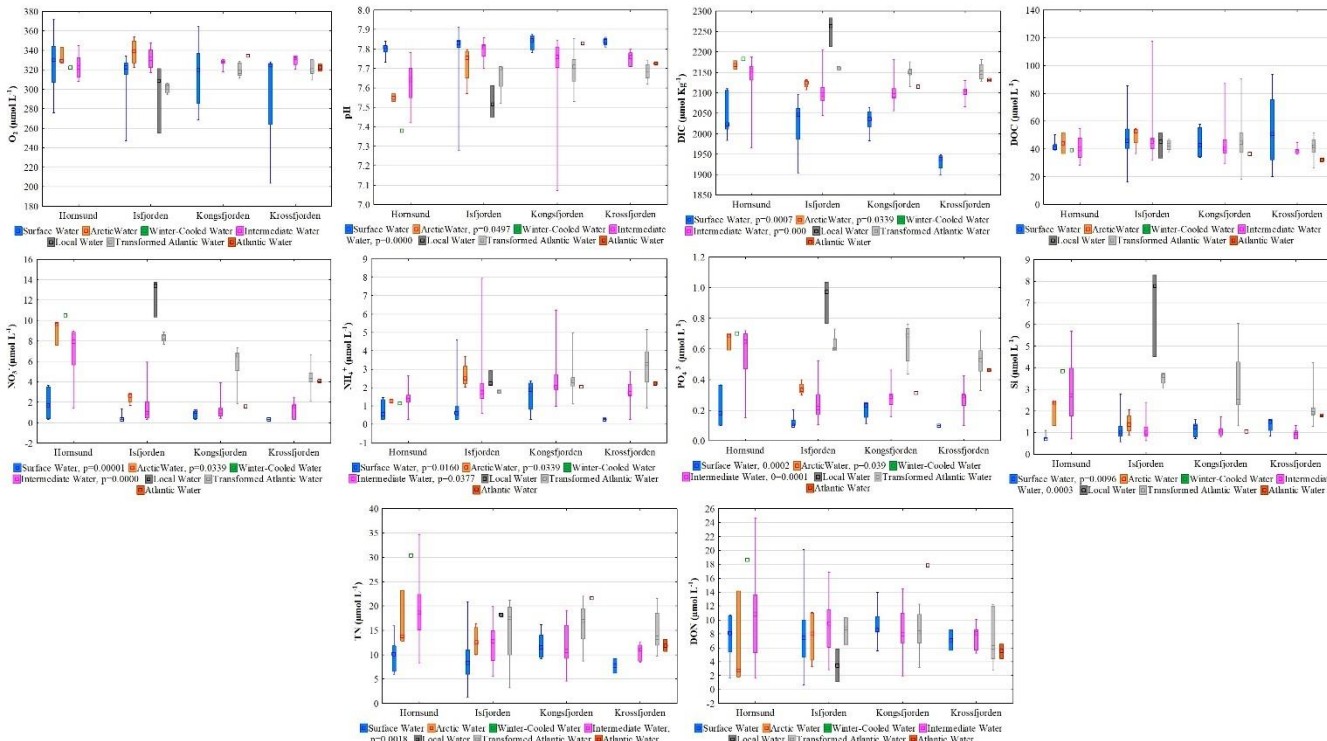

**Fig.5** Oxygen ($O_2$), pH, dissolved inorganic carbon (DIC), dissolved organic carbon (DOC), nitrate ($NO_3^-$), ammonium ($NH_4^+$), phosphate ($PO_4^{3-}$), dissolved silica (Si), total dissolved nitrogen (TN) and dissolved organic nitrogen (DON) in surface water (marked as blue), Arctic Water (marked orange), Winter-Cooled Water (marked green), Intermediate Water (marked pink),

Local Water (marked dark grey), Transformed Atlantic Water (marked light grey) and  Atlantic Water (marked red)  in Hornsund, Isfjorden, Kongsfjorden and Krossfjorden.



**Fig.6** Distribution of chloride ($Cl^-$), nitrate ($NO_3^-$), ammonium ($NH_4^+$), phosphate ($PO_4^{3-}$), dissolved silica (Si), dissolved inorganic carbon (DIC), dissolved organic carbon (DOC) in pore water in Hornsund (marked blue), Isfjorden (marked grey),
Kongsfjorden (marked red) and Krossfjorden (marked grey).



**Fig.7** Box plots of chloride (Cl⁻), nitrate (NO₃⁻), ammonium (NH₄⁺), phosphate (PO₄³⁻), dissolved silica (Si), dissolved inorganic carbon (DIC), dissolved organic carbon (DOC) in pore water (marked grey) and bottom water (marked orange) in

Hornsund, Isfjorden, Kongsfjorden and Krossfjorden.





**Author contributions**

SRS: Conceptualization, data interpretation, preparation of figures, investigation, writing–original draft, reviewing and editing. MB, WH, KK, AL, AS, BSz: Reviewing and editing.


**Competing interests**

The contact author has declared that none of the authors has any competing interests.

**Acknowledgements**

We thank our colleagues (Przemysław Makuch, Magdalena Diak, Marta Borecka, Katarzyna Koziorowska- Makuch, Fernando Aquado Gonzalo, Marcin Stokowski, Aleksandra Winogradow, Miłosz Grabowski, and Marek Zajączkowski) for sharing ideas and for help with field and laboratory work. We would like to acknowledge Laura Bromboszcz and Piotr Prusiński for their technical support. We would like to thank the captain and the crew of r/v OCEANIA.

**Financial support**

The authors declare financial support was received for the research, authorship, and/or publication of this article. The research leading to these results has received funding from the Norwegian Financial Mechanism 2014–2021 project no. 2019/34/H/ST10/00645 and 2019/34/H/ST10/00504. In addition, the present study is financed by statutory activities of the Institute of Oceanology Polish Academy of Science and National Science Centre project no. 2019/34/E/ST10/00167.

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
