# Peer review of "Distributions of in-situ parameters, dissolved (in)organic carbon, and nutrients in the water column and pore waters of Arctic fjords (Western Spitsbergen) during a melting season"

_Earth System Science Data, 2024_

## Author Comment (AC1)

**Response to Reviewer 1**

We wish to thank for the editor and the reviewers for their effort in handling of and commenting on our manuscript. We highly appreciate the insightful and helpful comments that helped to improve the hopefully final version of our manuscript.

**Comment 1**
L47: Is Atlantification not the same as sea ice losses?
**Answer to Comment 1**
Atlantification cause the sea ice loss. Warmer and saltier Atlantic water is extending its reach covering areas located more northward into the Arctic Ocean and as a result sea-ice is disappearing. The sentence has been changed to:
 *"In addition, Henley et al. (2020) indicated that, with ongoing sea ice losses due to Atlantification, the expected shift from more Arctic-like ice-impacted conditions to more Atlantic-like ice-free conditions is expected to increase nutrient availability and the duration of the vegetation period in the Arctic shelf region."*

**Comment 2**
L87: I think the authors mean a north-east to south west orientation.
**Answer to Comment 2**
The sentence is changed to:
*"Krossfjorden exhibits a north-east to south-west orientation, stretching approximately 30km in length and reaching widths from 3km to 6km."*

**Comment 3**
L111: "Seawater pH was measured 10ml of seawater was filtered." check text.
**Answer to Comment 3**
The sentence has been corrected and changed to:
*"10ml of seawater was filtered (cellulose acetate filters with a pore size of 0.45μm), frozen in a pre-cleaned high-density polyethene bottle and stored at -20°C for further nutrient analysis."*

**Comment 4**
Section 2.2.2 I see that for the porewater data there are quite some numbers missing, which might need some explanation in the methods section.
**Answer to Comment 4**
The explanation has been added to the description:
*"GEMAX and Nemisto gravity corers were used to collect up to approximately 40 cm long sediment cores, with inner diameter equal to 12 and 10cm, respectively. However, the retrieval of the cores in some locations was not possible due to the consolidated seafloor. Additionally, the pore water extracted from some sediment cores was insufficient to perform all analyses."*

**Comment 5**
L135: check size of letters.
**Answer to Comment 5**
The size of letters has been corrected.

**Comment 6**
Section 3.1. I find this section a bit hard to read because of all the abbreviations that for me are not very common. It might become better when figure 2 is plotted close to the text, however currently there is no reference to Fig. 2 in this section. Additionally, it might help to also plot the water masses as a "section plot" with potential T on the Y axis and distance on the x axis. I leave this up to the authors.
**Answer to Comment 6**
The reference to Fig.2 is added. The characterization of the water masses is provided in section 2.1 together with abbreviations, and we think that there is no need to repeat this information. We think that T-S diagram sufficiently represent the characterization of similarities and differences among the distribution of the water masses.

**Comment 7**
L154: Krossfjorden is marked yellow not grey.
**Answer to Comment 7**
Text has been improved:
*"...Krossfjorden (marked yellow)…"*

**Comment 8**
L163: (at 25°C) is mentioned after pH.
**Answer to Comment 8**
The methodology section has been improved and this particular sentence corrected.

**Comment 9**
L209. I find point two here a bit detailed compared to the other two. I would remove it. Also because there are more (micro)nutrients that can limit primary productivity such as iron and manganese.
**Answer to Comment 9**
Point number 2 has been changed to:
*" 2) the determination of C:N:P:Si ratios in different water masses and their comparison between fjords, as an assessment of the environmental controls and limiting factors for the primary production"*

**Comment 10**
Fig. 4 This figure is a bit unreadable. maybe it would hep if the authors plot three figures next to each other instead of 4.
**Answer to Comment 10**
The figure has been improved in accordance with the comment.

**Comment 11**
Fig 5. This figure is a bit unreadable. maybe it would hep if the authors plot three figures next to each other instead of 4.
**Answer to Comment 11**
The figure has been improved in accordance with the comment.

**Comment 12**
Fig. 6. There is an l missing after mmo and  umo in the unit of the figures.
**Answer to Comment 12**
The figure has been improved in accordance with the comment.

**Comment 13**
Krossfjorden 265 (marked grey). (actually marked yellow).
**Answer to Comment 13**
Changed.

---

## Author Comment (AC2)

**Response to Reviewer 2**

We wish to thank for the editor and the reviewers for their effort in handling of and commenting on our manuscript. We highly appreciate the insightful and helpful comments that helped to improve the hopefully final version of our manuscript.

**Comment 1**
The salinity and temperature data in Figure 2 seem to consist of binned data from the CTD casts. However, only salinity and temperature values from the discrete depths where water samples were collected are included in the data spreadsheet on Zenodo. The full dataset should be added.
**Answer to Comment 1**
Temperature, salinity and density used for preparation Fig.2 and classification of water masses were taken from CTD data from layers where discrete samples were collected. This information was added to the manuscript. In addition, data from the CTD was added to the database and its description was added in the manuscript.

**Comment 2**
The Cl$^-$ concentration data are not represented properly in the manuscript; the data spreadsheet contains water column [Cl$^-$] data, but this is not mentioned in the main text. Several of the water column [Cl$^-$] values are 0, which I suspect is an error (should they be NA?). There is also a 0 value in one of the pore water profiles, which is not included in Figure 6. If suspected outliers have been excluded from figures, I suggest making a note in the data spreadsheet.
**Answer to Comment 2**
Thank you. There should be blank (meaning that there is no data) instead of 0 value.
All the typos in the database have been corrected.

**Comment 3**
Statistical results are presented throughout the text, but in most cases it is not clear which test that have been used. A paragraph about statistics/calculations should be added to the methods section.
**Answer to Comment 3**
The section 2.3 Statistics and data analysis was added to the manuscript:
*"All statistical analyses were carried out using Statistica (Statistica 13) while the evaluation of the statistical significance was made using Kruskal-Wallis test. Figure 1 was prepared in Svalbard Map. Temperature-Salinity (TS) diagrams were made using python programing language while box plots were made by means of Statistica."*

**Comment 4**
L156-170: I don't really see the point of this comparison. This division of data mainly shows the effect of stratification. I suggest removing this piece of text; there is a clearer discussion about the distribution of parameters between fjords and water masses later in the text.
**Answer to Comment 4**

Indeed, the main driver in the distribution of water masses is stratification and freshening in surface water column. Still, we see the description of general patterns made in this section and visualized in Fig. 2 an important reference for next paragraphs. Thus, we would prefer to keep this unchanged.

**Comment 5**
L193-201: I disagree with this approach and do not see how it is of value to the manuscript. The pore water profiles occasionally display peaks close to the sediment-water, indicating production and possible release of the compound to the water. By taking an average of 5 cm, these details are erased. As such, the method does not give valuable information about whether the sediment is a source or a sink of these dissolved compounds. There is also no basis for using 5 cm rather than another value. It would be more relevant to point out any large-scale trends in profiles between fjords, and if there are individual profiles that stand out (and possible reasons as to why – situated by a river mouth, in a local depression, etc.).

**Answer to Comment 5**
Some of the collected cores were very shallow therefore to cover most of the sampling stations the concentrations from first 5 cm were used for the data interpretation. Then, bottom water and pore water results from each site were grouped into fjords. Fig. 7 show the median (minimal and maximal) concentrations for the entire fjord not particular site in order to have the general assessment if the sediments are a source or sink of chemicals and not to investigate the processes occurring in one particular site.

We agree with the Reviewer that to fully understand the fluxes through the sediment/water interface one would need to resolve/interpret the distribution of analytes in the top sediment layers of each single core separately and with higher resolution. However, it was not our intention in this data manuscript to quantify sediment/water fluxes, but to compare the general patterns occurring in different fjords and to highlight the potential of the pore waters dataset for further assessment and interpretation by data users. In our opinion, the integration over the first 5 cm and the obtained statistics in Fig. 7 is still a good approximation of whether on average sediments from different fjords may act as a source or sink of investigated constituents.

We included the Reviewer comment in the manuscript and changed the paragraph into:
*"To highlight the potential of the pore waters dataset for further assessment and interpretation by data users, the concentrations of investigated parameters in pore water up to 5 cm and the concentrations in bottom water were compared in Figure 7."*

**Comment 6**
Section 2.2.2: There are many values missing from the pore water data. Is this due to a lack of pore water, or caused by analytical issues?

**Answer to Comment 6**
The missing values are due to the lack of the material for analysis.
The explanation was sdded to section 2.2.2:
*"GEMAX and Nemisto gravity corers were used to collect up to approximately 40 cm long sediment cores, with inner diameter equal to 12 and 10cm, respectively. However, the retrieval of the cores in some locations was not possible due to the consolidated seafloor. Additionally, the pore water extracted from some sediment cores was insufficient to perform all analyses."*

**Comment 7**
L87-99: Invert the order of the paragraphs about Krossfjorden and Kongsfjorden, as the paragraph about Krossfjorden refers to information about Kongsfjorden.

**Answer to Comment 7**

The order has been improved.

**Comment 8**
L104: Add uncertainties for the temperature, salinity and oxygen measurements. Add oxygen sensor model.
**Answer to Comment 8**
Text improved and moved to section 2.2.1:
*"The accuracy of T, S and O2 equals to ±0.002 °C, ±1% and ±0.015%, respectively."*

**Comment 9**
L106-107: Move the information about the pH measurements to section 2.2.1.
**Answer to Comment 9**
Text improved and moved to section 2.2.1.

**Comment 10**
L107-108: Please add the inner diameters of the core liners.
**Answer to Comment 10**
The sentence is changed to:
*"GEMAX and Nemisto gravity corers were used to collect up to approximately 40 cm long sediment cores with inner diameter equal to 12 and 10cm, respectively."*

**Comment 11**
L111: Part of the sentence seems to be missing.
**Answer to Comment 11**
Corrected.

**Comment 12**
L114: Were the filters pre-combusted?
**Answer to Comment 12**
The filters were pre-combusted and information has been added to the text.

**Comment 13**
L118-119: Were the Rhizons inserted directly into the cores through pre-drilled holes in the core liner? Or were the cores sliced before pore water extraction?
**Answer to Comment 13**
The pore water was extracted through pre-drilled holes in the core liners via Rhizon® samplers directly after extracting the cores.
The information has been added to the text:
*"Pore water was extracted from sediments through pre-drilled holes in the core liners via Rhizon® samplers (Rhizosphere, diameter of 2.5mm, and mean pore size of 0.15µm) directly after extracting the cores."*

**Comment 14**
L121-122: Add volume and concentration of the $HgCl_2$ used.
**Answer to Comment 14**
"50µl" added to text.

**Comment 15**
Section 2.2.3: Most commonly the "nitrate" analysed is nitrate+nitrite. Is that the case here too, or is it only nitrite?
**Answer to Comment 15**
Samples were analyzed for nitrate using the SEAL AA500 AutoAnalyzer (Seal Analytical), as indicated in the text. During analysis we measured seperatly nitrite and nitrate + nitrite, from the difference we obtained nitrate and present the results.

**Comment 16**
L130: Add information about the uncertainty of the $Cl^-$ analysis.
**Answer to Comment 16**
Text added:
*"Chloride ($Cl^-$) was determined by titration (Mohr's Method) with precision of $0.1 mmol\ L^{-1}$."*

**Comment 17**
Section 3.1: Could you add a table with the defining characteristics for each water mass (salinity and temperature ranges, see Cottier et al. 2005)?
**Answer to Comment 17**
All the characteristics are presented in the cited references.

**Comment 18**
Figure 1: Please add extent indicators in the overview map, and annotate the panels according to the journal's requirements (e.g., a, b, c, d). Correct the coordinates of the individual fjord maps, they do not agree with the overview map. If the bathymetry of the fjords is available, this would make a valuable addition to the maps as it would give clearer information about the areas surrounding the stations.
**Answer to Comment 18**
The Figure is updated with adding annotations to each map and improving the coordinate in overview map. We feel that adding the bathymetry will make the figure too messy.

**Comment 19**
Figure 2: Since potential temperature is presented on the y axis, the isopycnals should be expressed as potential density anomalies rather than density anomalies (is this done?). The sign for the density anomaly is $\sigma_t$ (potential density anomaly: $\sigma_\theta$), not $\delta_0$. Do the colours really represent density, as the colours do not match the isopycnal lines? It would be more suitable to show e.g., depth with colour. Please ensure that the axes are the same for all panels to make comparisons easier. Annotate the panels according to the journal's requirements.
**Answer to Comment 19**
Yes, the isopycnals are indeed expressed as potential density anomalies rather than density anomalies. The Figure is updated with adding annotation to each plot and correcting the colors according to density.

**Comment 20**
Figure 3: I appreciate that it is difficult to represent large amounts of data in one figure, but these graphs are hard to read. The main purpose of this figure seems to be to show the differences between fjords and water masses. I suggest plotting the profiles in a grid of parameters versus

fjords (for the graphs to be large enough, this might require splitting the figure into two, e.g., parameters more and less influenced by biology). This would also allow data points to be coloured by water mass, which would help with the discussion about how the origin of the water affects its chemical composition. Furthermore, I would add lines between the datapoints in each profile.
**Answer to Comment 20**
The figure is improved.

**Comment 21**
Figure 4: I suggest removing this figure, see comment on L156-170.
**Answer to Comment 21**
Please see the answer to Comment 4 and accordingly we would like to leave in the manuscript this figure.

**Comment 22**
Figure 5: This figure is also very hard to read. Firstly, have one legend and placing it underneath all the graphs. Secondly, decrease the number of columns in the graph grid to three or even two, otherwise everything is too small to read. Thirdly, I suggest marking groups that are not significantly different with the same letters, rather than adding p values to the graph – it is currently not clear if the "significantly different" water mass is different to other water masses within the same fjord, or to the same water masses in other fjords, or both.
**Answer to Comment 22**
The figure is improved.

**Comment 23**
Figure 6: Like with figure 4, I suggest plotting the profiles in a grid of parameters versus fjords, and to add lines between the points in the individual profiles. The titles of the x axes are missing an l in 'mol'.
**Answer to Comment 23**
The figure is improved.

**Comment 24**
Figure 7: I think this figure can be removed, see comment on L193-201.
**Answer to Comment 24**
Please see the answer to Comment 5 and accordingly we would prefer to keep this figure in the manuscript.

---

## Author Comment (AC3)

Dear Dr. Sebastian van de Velde,

Thank you for your comments. Please find below respond to your comments and revised text.

**Comment 1**

1. Reviewer 2 has requested to add the oxygen sensor model (comment 8). 'Sea-Bird Scientific SBE 911 Plus CTD' is not the dissolved oxygen sensor. Please add the information about the oxygen sensor model.

**Answer to comment 1:**

The base model of the SBE 911 Plus CTD is equipped with the oxygen module SBE 43. The information is added: "... equipped with oxygen module SBE 43..."

**Comment 2**

2. It is unclear what the p-values in Figs 4, 5, and 7 refer to (what was compared with what?). Please clarify in the caption.

**Answer to comment 2:**

The p-value presents a significant difference in the median concentration of the parameter between the fjords.

The information is added: "**Fig.4** Box plots of a) temperature (T), b) salinity (S), c) pH, d) oxygen  $(O_2)$ , e) nitrate  $(NO_3^-)$ , f) ammonium  $(NH_4^+)$ , g) phosphate  $(PO_4^{3-})$ , h) dissolved silica (Si), i) dissolved inorganic carbon (DIC), j) dissolved organic carbon (DOC), k) total dissolved nitrogen (TN) and l) dissolved organic nitrogen (DON) in surface water (marked blue) and bottom water (marked orange) in Hornsund, Isfjorden, Kongsfjorden and Krossfjorden. The p-values indicate significant differences in the median concentration of the parameter between the investigated fjords."

"**Fig.5** a) Oxygen ( $O_2$ ), b) pH, c) nitrate ( $NO_3^-$ ), d) ammonium ( $NH_4^+$ ), e) total dissolved nitrogen (TN), f) dissolved inorganic carbon (DIC), g) dissolved organic carbon (DOC), h) dissolved organic nitrogen (DON), i) dissolved silica (Si), and j) phosphate ( $PO_4^{3-}$ ) in surface water (marked as blue), Arctic Water (marked orange), Winter-Cooled Water (marked green), Intermediate Water (marked pink), Local Water (marked dark grey), Transformed Atlantic Water (marked light grey) and Atlantic Water (marked red) in Hornsund, Isfjorden, Kongsfjorden and Krossfjorden. The p-values indicate significant differences in the median concentration of the parameter between the investigated fjords, presented only if statistically significant."

**"Fig.**7 Box plots of a) chloride (Cl-), b) nitrate (NO3-), c) ammonium (NH4+), d) phosphate (PO43-), e) dissolved silica (Si), f) dissolved inorganic carbon (DIC), and g) dissolved organic carbon (DOC) in pore water (marked grey) and bottom water (marked orange) in Hornsund, Isfjorden, Kongsfjorden and Krossfjorden. The p-values indicate significant differences in the median concentration of the parameter between the investigated fjords."

**Comment 3**

3. The text does not reflect your response to comment 12 of Reviewer 2. Please add the information about whether the filters were pre-combusted to the text.

**Answer to comment 3:**

The information is added.

**Comment 4**

4. I agree with Comment 17 of Reviewer 2 that adding a table with the defining characteristics for each water mass (salinity and temperature ranges, see Cottier et al. 2005) would help with readability. Please consider adding a table.

**Answer to comment 4:**

The Table was added to the text:

**"Table 1.** Salinity and temperature of various water masses in Hornsund, Isfjorden and Kongsfjorden-Krossfjorden. The classification was done based on Cottier et al., (2005), Nilsen et al., (2008) and Promińska et al., (2018).

|                                  | Hornsund
(Nilsen et al., 2008) |                 | Isfjorden
(Nilsen et al., 2008) |                 | Kongsfjorden-Krossfjorden
(Cottier et al. 2005) |                   |
|----------------------------------|-----------------------------------|-----------------|------------------------------------|-----------------|----------------------------------------------------|-------------------|
|                                  |                                   |                 |                                    |                 |                                                    |                   |
|                                  | Temperature (°C)                  | Salinity        | Temperature (°C)                   | Salinity        | Temperature (°C)                                   | Salinity          |
| Arctic Water (ArW)               | -1.5 > T > 2                      | 34 < S < 34.5*  | -1.5 > T > 1                       | 34.4 < S < 34.8 | -1.5 > T > 1                                       | 34.30 < S < 34.80 |
| Atlantic Water (AW)              | T > 3                             | S > 34.9        | T > 3                              | S > 34.9        | T > 3                                              | S > 34.65         |
| Intermediate Water (IW)          | T > 1                             | 34 < S < 34.7   | T > 1                              | 34 < S < 34.7   | T > 1                                              | 34.00 < S < 34.65 |
| Local water (LW)                 | T < 1                             |                 | T<1                                |                 | -1.5 > T > 1                                       | 34.30 < S < 34.85 |
| Surface Water (SW)               | T > 1                             | 34 < S          | T > 1                              | 34 < S          | T > 1                                              | S < 34            |
| Transformed Atlantic Water (TAW) | T > 1                             | 34.7 < S < 34.9 | T > 1                              | S > 34.7        | 1 > T > 3                                          | S >34.65          |
| Winter Cooled Water (WCW)        | T < -0.5                          | S > 34.4        | T < -0.5                           | S > 34.74       | T < - 0.5                                          | 34.40 < S < 35    |

\*Promińska et al. (2018)

**Comment 5:**

5. In Fig. 2, you have a,b,c,d on the figure, but you do not refer to it in the caption, please do so. **Answer to comment 5:**

Information added: "**Fig.1** a) Study Area including the general map of Spitsbergen, highlighting the depths of the fjords and the surrounding Svalbard shelf (a). The warm West Spitsbergen Current and cold East Spitsbergen Current are indicated by red and blue arrows, respectively (Vihtakari, 2022, 2020). Study sites located in b) Hornsund, c) Isfjorden, and d) Kongsfjorden and Krossfjorden are presented as black triangles.."

**Comments 6:**

6. in Figs 1, and 3-7, you do not use a, b, c, ... For clarity, it would help if you would consider adding a, b, ... on the figure and also in the captions.Answer to comment 6:Information added:

"Fig.1 a) Study Area including the general map of Spitsbergen, highlighting the depths of the fjords and the surrounding Svalbard shelf (a). The warm West Spitsbergen Current and cold East Spitsbergen Current are indicated by red and blue arrows, respectively (Vihtakari, 2022, 2020). Study sites located in b) Hornsund, c) Isfjorden, and d) Kongsfjorden and Krossfjorden are presented as black triangles."

**"Fig.2** The water masses distribution such as surface water (SW), Arctic Water (ArW), Winter-Cooled Water (WCW), Intermediate Water (IW), Local Water (LW), Transformed Atlantic Water (TAW) and Atlantic Water (AtW) in a) Hornsund, b) Isfjorden, c) Kongsfjorden and d) Krossfjorden."

"**Fig.3** Distribution of a) temperature (T), b) salinity (S), c) pH, d) oxygen ( $O_2$ ), e) nitrate ( $NO_3^-$ ), f) ammonium ( $NH_4^+$ ), g) phosphate ( $PO_4^{3^-}$ ), h) dissolved silica (Si), i) dissolved inorganic carbon (DIC), j) dissolved organic carbon (DOC), k) total dissolved nitrogen (TN) and l) dissolved organic nitrogen (DON) in Hornsund (marked blue), Isfjorden (marked grey), Kongsfjorden (marked red) and Krossfjorden (marked yellow)."

"**Fig.4** Box plots of a) temperature (T), b) salinity (S), c) pH, d) oxygen ( $O_2$ ), e) nitrate ( $NO_3^-$ ), f) ammonium ( $NH_4^+$ ), g) phosphate ( $PO_4^{3-}$ ), h) dissolved silica (Si), i) dissolved inorganic carbon (DIC), j) dissolved organic carbon (DOC), k) total dissolved nitrogen (TN) and l) dissolved organic nitrogen (DON) in surface water (marked blue) and bottom water (marked orange) in Hornsund, Isfjorden, Kongsfjorden and Krossfjorden. The p-values indicate significant differences in the median concentration of the parameter between the investigated fjords."

"**Fig.5** a) Oxygen ( $O_2$ ), b) pH, c) nitrate ( $NO_3^-$ ), d) ammonium ( $NH_4^+$ ), e) total dissolved nitrogen (TN), f) dissolved inorganic carbon (DIC), g) dissolved organic carbon (DOC), h) dissolved organic nitrogen (DON), i) dissolved silica (Si), and j) phosphate ( $PO_4^{3^-}$ ) in surface water (marked as blue), Arctic Water (marked orange), Winter-Cooled Water (marked green), Intermediate Water (marked pink), Local Water (marked dark grey), Transformed Atlantic Water (marked light grey) and Atlantic Water (marked red) in Hornsund, Isfjorden, Kongsfjorden and Krossfjorden. The p-values indicate significant differences in the median concentration of the parameter between the investigated fjords, presented only if statistically significant."

**"Fig.6** Distribution of a) chloride (Cl-), b) nitrate (NO3-), c) ammonium (NH4+),d) phosphate (PO43-), f) dissolved silica (Si), g) dissolved inorganic carbon (DIC), and h) dissolved organic carbon (DOC) in pore water in Hornsund (marked blue), Isfjorden (marked grey), Kongsfjorden (marked red) and Krossfjorden (marked yellow)."

**"Fig.**7 Box plots of a) chloride (Cl-), b) nitrate (NO3-), c) ammonium (NH4+), d) phosphate (PO43-), e) dissolved silica (Si), f) dissolved inorganic carbon (DIC), and g) dissolved organic carbon (DOC) in pore water (marked grey) and bottom water (marked orange) in Hornsund, Isfjorden, Kongsfjorden and Krossfjorden. The p-values indicate significant differences in the median concentration of the parameter between the investigated fjords."